# Changes in Maternal Heart Rate and Autonomic Regulation following the Antenatal Administration of Corticosteroids: A Secondary Analysis

**DOI:** 10.3390/jcm12020588

**Published:** 2023-01-11

**Authors:** Maretha Bester, Rohan Joshi, Joyce H. A. M. van Dooren, Massimo Mischi, Rik Vullings, Judith O. E. H. van Laar

**Affiliations:** 1Department of Electrical Engineering, Eindhoven University of Technology, 5612 AZ Eindhoven, The Netherlands; 2Patient Care and Monitoring, Philips Research, 5656 AE Eindhoven, The Netherlands; 3Faculty of Health, Medicine and Life Science, Maastricht University, 6200 MD Maastricht, The Netherlands; 4Department of Obstetrics and Gynecology, Máxima Medical Centrum, De Run 4600, 5504 DB Veldhoven, The Netherlands

**Keywords:** heart rate variability, maternal health, corticosteroids, heart rate, autonomic regulation, obstetrics, pregnancy, betamethasone, vagal tone

## Abstract

While the effect of antenatally administered corticosteroids on fetal heart rate (HR) and heart rate variability (HRV) is well established, little information is available on how these drugs affect maternal physiology. In this secondary analysis of a prospective, observational cohort study, we quantify how corticosteroids affect maternal HR and HRV, which serve as a proxy measure for autonomic regulation. Abdominal ECG measurements were recorded before and in the five days following the administration of betamethasone—a corticosteroid commonly used for fetal maturation—in 46 women with singleton pregnancies. Maternal HR and HRV were determined from these recordings and compared between these days. HRV was assessed with time- and frequency-domain features, as well as non-linear and complexity features. In the 24 h after betamethasone administration, maternal HR was significantly increased (*p* < 0.01) by approximately 10 beats per minute, while HRV features linked to parasympathetic activity and HR complexity were significantly decreased (*p* < 0.01 and *p* < 0.001, respectively). Within four days after the initial administration of betamethasone, HR decreases and HRV features increase again, indicating a diminishing effect of betamethasone a few days after administration. We conclude that betamethasone administration results in changes in maternal HR and HRV, despite the heterogeneity of the studied population. Therefore, its recent administration should be considered when evaluating these cardiovascular metrics.

## 1. Introduction

The use of antenatally administered corticosteroids has had a profound impact on the survival rates and outcomes of prematurely born neonates [1,2]. Therefore, their administration in cases of anticipated preterm birth or cesarean section is standard clinical practice. Still, many uncertainties remain regarding the perinatal use of corticosteroids. These uncertainties have been widely investigated and debated in the literature, such as the prudence and timing of administering a second course as well as the impact of in utero exposure to synthetic corticosteroids on the child in the long term [3]. However, a potential effect that is rarely investigated or addressed is the influence of these maternally administered corticosteroids on the cardiac and autonomic nervous systems (ANS) of the mother [4,5].

Research suggests that glucocorticoids, the class of steroids to which corticosteroids belong [6], may act as a sympathetic cardiovascular stimulus that influences heart rate (HR) [7]. Changes in maternal vital signs inform clinicians of the mother’s well-being and consequently play a role in clinical decision-making [8,9]. Therefore, quantifying the potential effect of corticosteroids on maternal HR is necessary.

Additionally, since pregnancy is accompanied by substantial changes in the maternal cardiovascular system and ANS, drugs administered during pregnancy that might affect these systems need to be carefully studied [10]. Further compounding this need is the fact that many women who receive corticosteroids also have pregnancy complications, such as hypertensive disorders of pregnancy. These complications not only alter maternal cardiac and autonomic function but also increase the long-term risk for cardiovascular diseases [11,12]. Subsequently, it is important to investigate whether administering these synthetic corticosteroids—which incidentally also increase the risk of cardiovascular disease with long-term use [7]—further alters maternal autonomic regulation.

Understanding how administering corticosteroids alters maternal autonomic regulation is not only physiologically important but also clinically relevant. Owing to the association between maternal autonomic dysfunction and pregnancy complications [10,13], there is increasing interest in using maternal heart rate variability (HRV)—a proxy measure for the ANS [14]—as an obstetric screening tool [15,16,17]. However, using HRV to track maternal health, as well as appropriately interpreting HRV in women who’ve received corticosteroids, requires quantifying the potential effect of this routinely administered medication on maternal HRV [11,18].

Subsequently, in this study, we investigate the effects of betamethasone—the most commonly used corticosteroid in the antenatal period—on the maternal system. The objectives of our analysis are twofold. First, we investigate whether antenatally administering betamethasone alters maternal HR in a clinically relevant manner. Second, we determine whether receiving this drug results in altered maternal autonomic regulation, as assessed via HRV.

## 2. Materials and Methods

We perform a secondary analysis of a dataset of abdominal ECG measurements collected during a longitudinal, observational cohort study (March 2013 to July 2016) at the Máxima Medical Centre (Máxima MC), a tertiary care teaching hospital in Veldhoven, The Netherlands. The primary purpose of the data collection was to quantify the effect of betamethasone on fetal HRV; results from this analysis were previously published [19,20]. The original study was approved by the institutional review board at Máxima MC in Veldhoven, The Netherlands (NL43294.015.13), which granted a waiver for this secondary analysis (N21.008).

### 2.1. Study Population

Women with singleton pregnancies who were admitted to the Obstetric High Care Unit at Máxima MC with a risk for preterm delivery were recruited for this study. The aim was to include at least 50 patients. Those who received betamethasone (Celestone Chrondose^®^, Schering AG, Berlin, Germany; 2 doses of 12 mg intramuscularly, 24 h apart) as part of their standard clinical care were eligible to participate in the study. Co-administration of medications was allowed since this was part of the standard treatment protocol. Nifedipine was administered as a tocolytic to attenuate contractions as needed in cases of threatened preterm labor, at times in conjunction with indomethacin when contractions persisted while betamethasone treatment had not yet been completed. Antibiotics (erythromycin 250 mg, 4 times daily for 10 days) were administered to patients with preterm rupture of membranes to prevent infection. Furthermore, women with preeclampsia typically received antihypertensive drugs (specifically, methyldopa or labetalol). Patients under 18 years of age were not eligible for participation. Metadata collected from participants were indications for betamethasone administration, medications administered during the study period, parity, body mass index (BMI), gestational age at inclusion, and general and obstetric medical history [19].

### 2.2. Measurements

Repeated abdominal ECG measurements—from which maternal HR and HRV can be determined—were performed over several days. Measurements, which were approximately 30 min in duration, were recorded while the patient was lying in a semi-recumbent position. Ideally, the first measurement took place before betamethasone was administered; this timestamp was defined as *day 0*. Thereafter, measurements were taken over the next five days at approximately 24 h intervals, i.e., at 24 h after (*day 1*), 48 h (*day 2*), 72 h (*day 3*), 96 h (*day 4*), and finally 120 h (*day 5*) after the first betamethasone injection. Each measurement was performed between 20 and 28 h after its preceding measurement to reduce the impact of diurnal rhythms. No measurements were performed between midnight and 07h00.

### 2.3. Analysis

As previously noted, this study is a secondary analysis of a dataset originally collected to assess the effect of betamethasone on fetal HRV. This dataset was originally collected to compare the progressing changes in fetal HRV after the administration of betamethasone against a reference measurement. Ideally, this reference measurement would occur before the administration of betamethasone. However, Máxima MC is a tertiary care hospital to which many patients are transferred after an initial assessment at their primary care hospital. Consequently, women often have already received their first injection of betamethasone before arriving at Máxima MC. Additionally, evidence from the literature indicates that the effect of betamethasone on fetal HRV wears off within four days of the first injection [21]. Therefore, theoretically, *day 4* or *day 5* could serve as the reference measurement in lieu of *day 0* (i.e., the measurement taken before betamethasone is administered) when investigating the effect of the drug on fetal HRV. Therefore, the original protocol stipulated that measurements would be collected for five days following the first betamethasone injection [19].

However, limited information is available on the expected effect of betamethasone on maternal HRV. Therefore, we make no assumptions on the duration of the effect of this drug on the maternal HRV and define no reference measurement in addition to *day 0*. Rather, we track the potentially transient effect of betamethasone on maternal HRV over the six days of measurements. We assess the overall trend in maternal HR and HRV as well as compare them individually between all days. Owing to the explorative nature of the analysis, we include all participants, regardless of their number of measurements.

### 2.4. Outcome Measures

The outcomes of interest are maternal HR and maternal HRV. The latter is quantified by HRV features from the time domain and frequency domain, as well as features describing the complexity or non-linearity in the HR signal. The features used to capture HRV are detailed in Section 2.6.

### 2.5. Data Acquisition and Signal Pre-Processing

The multichannel abdominal ECG recordings were acquired with one of two non-invasive electrophysiological monitoring devices, namely the Nemo (Nemo Healthcare BV, Eindhoven, The Netherlands) at a sampling rate of 500 Hz and the Porti system (Twente Medical Systems International B.V., Enschede, The Netherlands) at a sampling rate of 512 Hz. A 4th-order Butterworth bandpass filter of 1 to 70 Hz was applied to the recordings to suppress artifacts and out-of-band noise, followed by a notch filter at 50 Hz, which suppressed powerline interference. Next, a fixed linear combination of the various abdominal channels was applied to enhance maternal QRS peaks [22], and hereafter a previously detailed peak detector was used to detect the maternal R-peaks [23,24]. Once these peaks were detected, signals representing the RR-intervals could be generated. These signals were further pre-processed to reduce the impact of potential noise and erroneous beats. Physiologically improbable RR-intervals (shorter than 0.4 s or longer than 2 s) or those with too large a change between consequent RR-intervals (i.e., a change of more than 20%) were rejected [25,26,27]. For the calculation of HRV features that relate to the frequency domain or complexity, a continuous time series is needed, and subsequently, missing RR-intervals are replaced with cubic spline interpolation when calculating these features. Recordings were excluded from the analysis if more than 10% of RR-intervals were rejected.

### 2.6. HR and HRV Analysis

First, the HR is determined by taking the average of the RR intervals and converting this to beats per minute (bpm). Thereafter, HRV features are calculated. Each feature is calculated over the entire recording; for the HRV features in the frequency domain, the average is taken after the features are calculated on 5-min segments from the recording with a 50% overlap between segments. The set of time domain features comprises the standard deviation of the RR intervals (SDNN), the root mean square of the successive differences of the RR intervals (RMSSD), and the percentage of consecutive RR intervals that differ by more than 50 ms (pNN50). The latter two capture parasympathetic activity (of which the vagus nerve is the main component influencing HR), since such short-term variations are mediated by the parasympathetic nervous system, while SDNN represents the overall HRV [14].

HRV can also be assessed in the frequency domain, and subsequently, we determine the following features: the power in the high-frequency band of 0.15–0.40 Hz (HF), the power in the low-frequency band of 0.04–0.15 Hz, and the ratio between the two (LF/HF). HF captures mainly parasympathetic activity, while LF is influenced by both branches of the ANS (i.e., both parasympathetic and sympathetic activity). LF/HF informs on the balance between the two branches by capturing what is referred to as the sympathovagal balance [14,28].

Additionally, we also assess the non-linearity and complexity of the signal representing the RR-intervals. We use a popular geometrical method to evaluate HRV dynamics, namely a Poincaré plot, in which each RR-interval is plotted against its predecessor to form a scatter plot. An ellipse is then fitted to the plot, from which two standard descriptors (SD1 and SD2) are calculated to represent the short- and long-term HRV, respectively. These are presented as a ratio (SD1/SD2) that informs on the relationship between long- and short-term variability, which—similarly to LF/HF—offers a window into sympathovagal balance [14,29]. Furthermore, we assess the complexity of the signal representing the RR-intervals with two methods: Sample entropy (SampEn) and detrended fluctuation analysis (DFA) [30,31]. SampEn quantifies the conditional probability that two epochs that are similar within a tolerance r for a window length m will remain similar when including the next data point (i.e., the next RR-interval) [31,32]. The parameters *m* and *r* were set to 2 and 0.2 times the standard deviation of the RR-intervals [31]. Lower SampEn indicates a more regular and predictable time series [14]. Concerning DFA, this method is used to quantify the self-similarity of RR-intervals over time. In a healthy HR pattern, we expect that certain trends will repeat over different timescales; subsequently, the signal that represents the RR-intervals should be neither fully predictable nor completely random, but rather somewhere in between. To capture this characteristic, we calculate the short-term fractal scaling exponent of the DFA, namely α_1_, which represents the correlation in the RR-signal over 4–16 heartbeats [30]. A result of α_1_ = 1 suggests a high level of complexity. As α_1_ decreases, the complexity level decreases [30,33].

### 2.7. Statistical Analysis

Since the data are not normally distributed, non-parametric analyses were performed. We performed a Kruskal–Wallis test, which was used for comparing multiple independent samples, to ascertain whether significant changes in maternal HR and HRV occurred over the six days. Furthermore, we use Dunn’s post hoc test with Bonferroni correction to test for differences between the days. A value of *p* < 0.05 is seen as significant. Results are presented as boxplots with the appropriate statistics added to the plots.

## 3. Results

A total of 68 women were initially enrolled. Three participants withdrew from the study, while five had no measurements available. Seven participants ultimately did not receive betamethasone and were therefore excluded. Two more women were excluded due to known cardiac arrhythmias, and two more were excluded due to delivering and being discharged immediately after inclusion, respectively. Finally, the data from three women were excluded from the analysis due to their having no measurements taken at the correct times with regards to the betamethasone injection, resulting in 46 women being included. Of these inclusions, 31 had measurements from the Nemo device, 13 from the Porti system, and two had measurements from both monitoring systems. The characteristics of those included can be found in Table 1.

For these 46 women, a total of 219 recordings were available, of which 21 were rejected for having more than 10% unreliable RR-intervals. A further 15 measurements were taken outside the timeframe specified by the study protocol (i.e., within 20–28 h after the previous measurement) and were consequently excluded from the analysis. Subsequently, 183 recordings are included, of which there are 13 on *day 0*, 59 on *day 1*, 39 on *day 2*, 31 on *day 3*, 24 on *day 4*, and 17 on *day 5*. In some cases, women had two recordings available per day; both recordings were then incorporated. Note that the recordings for *day 1* could be either before or after the second injection of betamethasone, which is given 24 h after the first injection. We compared these two sets of recordings (i.e., recordings taken shortly before and shortly after the second betamethasone injection) and found neither apparent nor statistically significant differences between the two sets; subsequently, all these recordings were included as *day 1*.

### 3.1. Maternal HR

Maternal HR changes significantly (*p* < 0.01) following the administration of betamethasone (Figure 1). Compared to the pre-betamethasone measurement (*day 0*), HR is significantly increased by about 10 bpm 24 h after the first injection and reduces significantly again on *day 3* to a similar level as pre-betamethasone administration. No further significant differences are present after *day 3*.

To confirm the increase in HR observed from pre-betamethasone administration (*day 0*) to post-administration (*day 1*), we also considered a subanalysis with only participants who had measurements on both of these days (n = 12). Correspondingly, we observed a similar increase from a median of 81.7 bpm (interquartile range: 77.8–83.5 bpm) to a median of 88.4 bpm (interquartile range: 85.0–91.8 bpm).

### 3.2. Maternal HRV

Figure 2 details the changes in maternal HRV features spanning *day 0* to *day 5*. Overall, the most noticeable change in the HRV features can be observed on *day 1* in comparison to the preceding or following days. Note that all statistically significant relationships seen in Figure 2 are in comparison to *day 1*.

While no significant differences are seen in SDNN and LF, all features linked to parasympathetic activity (RMSSD, pNN50, and HF) are significantly decreased (*p* < 0.01). Furthermore, all three features also show significant (*p* < 0.05) differences between both *day 0* and *day 1*, as well as between *day 1* and *day 3*.

LF/HF and SD1/SD2 (representing sympathovagal balance) change significantly over the five days (*p* < 0.01 and *p* < 0.001, respectively). Furthermore, LF/HF increases and SD1/SD2 decreases on *day 1* (both signaling reduced vagal control) compared to day zero. Both thereafter revert to values similar to the pre-betamethasone period via significant changes between *day 1* and *days 2* and *3*.

Likewise, both features that are linked to complexity (SampEn and α_1_ from DFA) show significant changes (*p* < 0.01 and *p* < 0.001, respectively) over the study period. The drop in SampEn and the increase in α_1_ on *day 1* indicate that there is a decrease in complexity following the first injection of betamethasone. In both cases, the complexity increases again on *day 2* and *day 3*, with statistically significant relationships present between *day 1* and *day 2* as well as *day 1* and *day 3* (*p* < 0.01). In the case of α_1_, *day 4* is also significantly reduced compared to *day 1* (*p* < 0.05).

## 4. Discussion

This study provides evidence that antenatally administered corticosteroids (specifically, betamethasone) significantly influence maternal physiology. Maternal HR (Figure 1) is increased by about 10 bpm within 24 h after the first betamethasone injection before returning to levels similar to those pre-administration. Parasympathetic activity (Figure 2) significantly decreases after betamethasone administration (*day 1*) before stabilizing (*day 2* or *day 3*). Features representing sympathovagal balance (i.e., LF/HF and SD1/SD2) also exhibit decreased vagal control on *day 1*, while HR complexity is also significantly decreased on *day 1* (Figure 2). Finally, for all features, the most notable change is that on *day 1* compared to the pre-betamethasone period (*day 0*) and the two- or three-days following *day 1*. Consequently, any significant effects of betamethasone on maternal HR and HRV likely wear off by *day 3* or *day 4*.

This is the first study, to our knowledge, that addresses the effect of antenatally administered betamethasone on the maternal cardiac and ANS. A major strength is the longitudinal nature of the dataset analyzed. Assessing measurements ranging from pre-betamethasone administration to 120 h thereafter allows for tracking the effect of corticosteroids over several days rather than merely assessing the immediate effect. This longitudinal analysis also allows us to observe when the effect of betamethasone appears to be mitigated.

Still, more nuanced effects of betamethasone are perhaps obscured by the unpaired nature of our analysis, the small number of participants, the differing number of measurements available per day, the largely unavoidable co-administration of other medications such as nifedipine, and the heterogeneity of our dataset in terms of complications, age, parity, etc. Furthermore, the lack of information on blood pressure (BP)—which is likely to also be affected by corticosteroids [34]—limits physiological interpretation. Yet, despite these limitations, we still observe large statistically significant changes in our outcome measures, increasing our confidence that betamethasone indeed affects maternal HR and HRV. Furthermore, the heterogeneity of our study group reflects the typical characteristics of the population that receives corticosteroids during pregnancy, thereby increasing the clinical relevance of these results.

Changes in the maternal heart rhythm are relevant for clinical decision-making [8,9]. In a recent review on assessing and interpreting maternal bradycardia and tachycardia, the administration of medications such as beta-blockers is listed as a potential cause of bradycardia. However, for tachycardia (defined in the review as maternal HR > 100 bpm), no medications are considered to be potential triggers for an elevated maternal HR [8]. Yet, we find that approximately 24 h after betamethasone administration, maternal HR is elevated by about 10 bpm (Figure 1). Since this may result in perceived tachycardia in women with a high baseline HR (as is quite common during pregnancy [35]), we believe clinicians should consider whether corticosteroids have recently been administered when evaluating an elevated maternal HR.

Along with these changes to maternal HR, we observe significant changes in several HRV features (Figure 2). Significant changes in RMSSD, pNN50, and HF—as well as LF/HF and SD1/SD2—suggest reduced parasympathetic activity after betamethasone administration (Figure 2), while the lack of change in LF suggests stable sympathetic activity (Figure 2). The decreased parasympathetic activity implies reduced vagal control of the heart, which is also reflected in the increased maternal HR (Figure 1). It should be noted that an increased HR typically results in lower variability in the HR signal, translating to generally lower HRV [36]. However, while the increased HR likely contributes to the decreased HRV features, SDNN—which is a measure of overall variability—is not significantly reduced. This suggests that additional physiological mechanisms likely also play a role in reducing the affected HRV features.

All significant changes in maternal HR and HRV seem to be mitigated by *day 3* or *day 4*; a similar timeline is seen in the case of fetal HRV [19,20,21]. Therefore, investigations into maternal HRV in complicated pregnancies should take care to perform measurements either preceding or at least four days following betamethasone administration.

Presumably, these changes in our outcome measures may also be a result of either the stress of hospitalization due to a complicated pregnancy or the consequence of the co-administration of other routine obstetric medications. However, the transient nature of the changes observed on *day 1*, along with the apparent return to normal on *day 3*, suggests that these effects result from betamethasone administration. Furthermore, the most commonly co-administered medication is nifedipine, and researchers have found that while nifedipine provokes reflex tachycardia, HR returns to baseline within two hours [37]. Therefore, it is unlikely to be the main driver of the observed changes.

Still, the effect of nifedipine on maternal HR and HRV is not well understood. Subsequently, to ensure that nifedipine does not markedly affect the results of our analysis, we also performed a sub-analysis with the 21 participants who did not receive this drug (see Figure A1 and Figure A2 in Appendix A). In these figures, we observed the same trends in outcome measures, albeit with fewer statistically significant changes, seeing as the sample size was more than halved. Additionally, we repeated this sub-analysis with the 41 participants who did not receive anti-hypertensive drugs (see Figure A3 and Figure A4 in Appendix A). While only five participants received these drugs, anti-hypertensives are known to affect HRV. Similarly, there was little noticeable change in the trends observed in these figures, increasing our confidence that the changes we observe are primarily due to betamethasone administration.

No work has been published on how betamethasone might influence maternal HR and HRV apart from a case report detailing maternal bradycardia after betamethasone administration (a rare but known side effect [38]) and a small animal study assessing maternal HR [39]. The latter compares the HR between four pregnant baboons who receive betamethasone and five controls; no significant changes were observed between these two groups during the 72-h study period. Subsequently, little guidance is available from the literature on the potential mechanism behind the changes we observe in maternal HR and HRV after betamethasone administration.

We hypothesize that these changes in maternal physiology relate to the effect of betamethasone on the hypothalamus-pituitary-adrenal axis (HPA-axis), a major neuroendocrine system that plays an important role in the long-term stress response via cortisol secretion [6]. Increased cortisol, which is an endogenous glucocorticoid, invokes the cardiovascular stress response, leading to increased BP and cardiac output [34]. When betamethasone—an exogenous glucocorticoid—enters the body, an increased level of cortisol is detected, and the HPA axis is suppressed via a negative feedback mechanism [40,41,42]. The biological half-life of betamethasone, i.e., the period during which the HPA-axis is suppressed and cortisol is correspondingly increased, is 36 to 59 h [43,44], suggesting that the effects of betamethasone should start to wane around *day 3* (i.e., approximately 48 h after the second injection). Researchers have demonstrated a negative relationship between cortisol levels and vagal tone [45,46,47]; correspondingly, in Figure 2, features linked to vagal activity (RMSSD, pNN50, and HF) are decreased on *days 1* and *2* before normalizing around *day 3*. Furthermore, HR is increased in this period, likely due to the decreased vagal control (which acts as a “brake” on the HR) and the increased cardiovascular stress response. While the stress response is typically associated with sympathetic activation, our results indicate that sympathetic activity is not significantly altered after betamethasone administration (LF from Figure 2). Conversely, one study has shown that administering exogenous glucocorticoids daily for a week inhibits baseline sympathetic outflow [48]. Although differences in study setup prohibit direct comparison between our results and theirs, it is evident that the relationship between the HPA axis and the ANS is still poorly understood, as are the pathways by which cortisol invokes the cardiovascular stress reflex [49,50].

Interestingly, the opposite response is observed in the fetus, where HR decreases and HRV increases in the first 24 h [19]. Researchers hypothesize that it is due to the activation of the fetal baroreflex in response to the increase in fetal BP, which lowers HR and increases HRV [19]. It should also be noted that the half-life of betamethasone in the fetus is double that of the mother [51]. Subsequently, even if similar effects occur in both mother and fetus, these would likely be at different timescales. Further, considering that we only have spot measurements every 24 h, it is not possible to conclude why the responses in the mother and fetus seem to differ.

However, regardless of the physiological mechanisms at play, we find that betamethasone influences autonomic activity in women with pregnancy complications. The next step would be to confirm these findings with a paired analysis from a prospective study dedicated to investigating the effect of betamethasone on maternal physiology [18]. Future studies should investigate the potential impact of betamethasone on additional markers of maternal health, for example, BP. This is particularly important in pregnant women with already elevated BP, such as those with hypertensive disorders of pregnancy, as corticosteroids are likely to further increase BP [34].

## 5. Conclusions

Antenatally administered betamethasone increases maternal HR, and therefore its recent administration should be considered when evaluating maternal tachycardia. Furthermore, betamethasone alters autonomic regulation in these women, who already have dysfunctional autonomic regulation owing to their pregnancy complications. Further investigation is necessary to ascertain whether this has an impact on the health of the mother and whether the changes in maternal physiology observed have additional clinical significance.

## Figures and Tables

**Figure 1 jcm-12-00588-f001:**
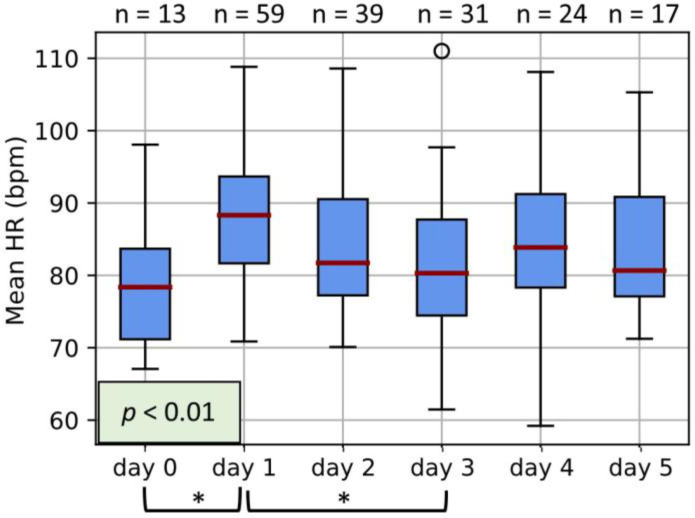
Boxplots of the HR of participants on *days 0* to *5*. Boxplots represent the median, interquartile range, and interdecile range of the values. *Day 0* represents measurements taken before the first injection of Betamethasone, while *day 1* represents 24 h after the first injection, *day 2* represents 48 h after the first injection, etc. * represents a statistically significant finding with *p* < 0.05. The number of measurements incorporated into each day is displayed at the top of the graph.

**Figure 2 jcm-12-00588-f002:**
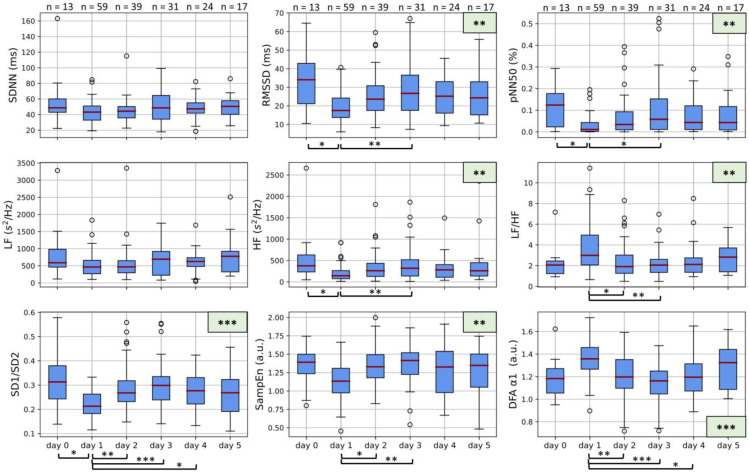
Boxplots of maternal HRV on *days 0* to *5*. Boxplots represent the median, interquartile range, and interdecile range of the values. *Day 0* represents measurements taken before the first injection of Betamethasone, while *day 1* represents 24 h after the first injection, *day 2* represents 48 h after the first injection, etc. * represents a statistically significant finding with *p* < 0.05, while ** presents *p* < 0.01 and *** represents *p* < 0.001. The number of measurements incorporated into each *day* is displayed at the top of the graph.

**Table 1 jcm-12-00588-t001:** Patient characteristics presented as occurrence or median and interquartile range, as appropriate.

Characteristic	
Indication for betamethasone (no. of participants)	
Threatened preterm labor	18
Vaginal bleeding	9
Preterm rupture of membranes	12
Preeclampsia	2
HELLP	2
Fetal intra-uterine growth restriction	3
Gestational age at inclusion (weeks + days)	29 weeks 2 days (26 weeks–31 weeks 2 days)
BMI	24.4 (21.9–29.3) kg/m^2^
Nulliparous	41.3%
Co-administration of other medications (no. of participants)	
Antibiotics (erythromycin or Augmentin)	13
Nifedipine	25
Anti-hypertensive (methyldopa or labetalol)	5
Magnesium sulfate	5
Atosiban	3

## Data Availability

The data that support the findings of this study are available from the corresponding author upon reasonable request.

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
