# Peer review of "Changes in Maternal Heart Rate and Autonomic Regulation following the Antenatal Administration of Corticosteroids: A Secondary Analysis"

_jcm, 2023, doi:10.3390/jcm12020588_

Round 1

Reviewer 1 Report

Bester et al investigate heart rate and heart rate variability changes in 46 women receiving antenatal corticosteroids using 30 minute abdominal ECG recordings.  They found an increase in heart rate and reduction in heart rate variability consistent with reduced parasympathetic tone.  The findings are consistent with known biological effects of steroids. 

Major comments

1.     Not all patients had baseline measurements

2.     There was no control group

3.     A number of patients received nifedipine, methyldopa and labetalol and the impact of these medications is not reported

4.     The clinical significance of these findings is unknown

Author Response

Reviewer 1

Bester et al investigate heart rate and heart rate variability changes in 46 women receiving antenatal corticosteroids using 30 minute abdominal ECG recordings.  They found an increase in heart rate and reduction in heart rate variability consistent with reduced parasympathetic tone.  The findings are consistent with known biological effects of steroids.

Thank you for your comments. Please find them addressed pointwise below. Track Changes has been used to facilitate finding changes made to the manuscript. Furthermore, when sections of the manuscript are mentioned as part of a response to a comment, then these will be highlighted in the manuscript. The line numbers mentioned below correspond to when All Markup is shown in the revised Word document.

Major comments

  1. Not all patients had baseline measurements

This is correct. For this reason, we did not perform a paired analysis. We have added this more clearly in the Statistical analysis section:

Line 198 - 199: ‘We perform a Kruskal-Wallis test, which is used for comparing multiple independent samples, to ascertain whether significant changes in maternal HR and HRV occur over the six days.

Additionally, we acknowledge in our limitations that a paired analysis would have been preferable.

Line 293 - 294: ‘Still, more nuanced effects of betamethasone are perhaps obscured by the unpaired nature of our analysis…

Furthermore, we have added to the future work section that performing a paired analysis would be the logical next step to confirm the findings of this investigation.

Lines 389 - 391: ‘The next step would be to confirm these findings with a paired analysis from a prospective study dedicated to investigating the effect of betamethasone on maternal physiology [18].’

  1. There was no control group

You are right, this is an observational cohort study so there is no control group. We have made this more clear in the revised manuscript. Please see:

Line 16: ‘In this secondary analysis of a prospective, observational cohort study…

Line 73 - 74: ‘We perform a secondary analysis of a dataset of abdominal ECG measurements collected during a longitudinal, observational cohort study…

  1. A number of patients received nifedipine, methyldopa and labetalol and the impact of these medications is not reported.

Thank you for your observation. Several patients receive these medications and it’s indeed important to clearly address these. Subsequently, we performed a subanalysis to investigate the effect of nifedipine and anti-hypertensives on our results. We have clarified the results of these in our Discussion. Please see lines 329 to 348, as well as Appendix A (pages 10 – 12) to which these paragraphs refer. Additionally, the number of participants who received these medications are added to Table 1 on page 5.

  1. The clinical significance of these findings is unknown

We agree with the reviewer that this significance is still unknown and should be studied. Changes in the maternal heart rhythm are relevant for clinical decision-making, as we discuss in lines 304 to 313. As this is the first study to demonstrate that there is indeed an effect on maternal HR and HRV in response to betamethasone administration, we did not investigate further clinical significance. Now that it has been demonstrated that corticosteroids indeed affect maternal physiology, this lays the foundation for future studies to investigate further potential clinical significance. Please see the following added:

See lines 402 - 405: ‘Further investigation is necessary to ascertain whether this has an impact on the future health of the mother and whether the changes in maternal physiology observed have additional clinical significance.’

Reviewer 2 Report

Despite the clearly noted limitations of the study, the present manuscript consists of a well written and  detailed presentation of the research perfomed. Especially discussion section , where authors attempt to give a point-to analysis and reasoning of the results.

Author Response

Reviewer 2

Despite the clearly noted limitations of the study, the present manuscript consists of a well written and  detailed presentation of the research perfomed. Especially discussion section , where authors attempt to give a point-to analysis and reasoning of the results.

Thank you for your comments.

Reviewer 3 Report

This is a useful study of cardiovascular changes in pregnant women treated with betamethasone to increase fetal maturity. A weakness is the transverse design of the study. Furthermore, I miss blood pressure data.

ABSTRACT

“The effects of betamethasone typically diminished within four days after initial administration” The meaning of this sentence is not clear. Data showed a significant difference between day 1 compared to day 0 or day 3, but a trend over the first 4 days is not clear.

METHODS

RESULTS

Fig 1: It might be useful to assess if the difference between D0 and D1 can be observed when the analysis is restricted to those who had a measurement at both instants.

“all features linked to parasympathetic activity (RMSSD, pNN50, and HF) are significantly altered” – possibly it is clearer to describe the direction of the change = decreased.

“The drop in SampEn……” : I thought that higher heart rate is generally associated with a reduction in variability of the signal. Are the changes that you observe larger than you would expect just from the HR shift?

DISCUSSION

You might describe more clearly that the difference of the numbers of registrations between D0 and D1 and following days could introduce bias. Ideally the study should have a repeated measurements design, but I fully understand that general clinical practice precludes this.

Line 331: There is some information on glucocortosteoids (but not on betamethasone) and ANS : e.g Clin Pharmacol Ther 1995;58:90-8, which shows (approximately) comparable data to this study

“Future studies should further investigate the potential long-term impact of these effects, as women with complicated pregnancies already have abnormal autonomic regulation and an increased risk for cardiovascular disease later in life.” I doubt if it is realistic to expect a long-term effect of a single course of betamethasone in adult women. Furthermore, the increased risk of later cardiovascular disease depends on the same risk factors that cause the obstetric pathology.

Author Response

Reviewer 3

This is a useful study of cardiovascular changes in pregnant women treated with betamethasone to increase fetal maturity.

Thank you for your review. We address your additional comments pointwise below. Track Changes has been used to facilitate finding changes made to the manuscript. Furthermore, when sections of the manuscript are mentioned as part of a response to a comment, then these will be highlighted in the manuscript. The line numbers mentioned below correspond to when All Markup is shown in the revised Word document.

A weakness is the transverse design of the study. Furthermore, I miss blood pressure data.

In the Discussion, we have further emphasized these two limitations.

Lines 297 - 298: ‘Furthermore, the lack of information on blood pressure (BP) – which is likely to also be affected by corticosteroids [34] – limits physiological interpretation.

Lines 389 - 391: ‘The next step would be to confirm these findings with a paired analysis from a prospective study dedicated to investigating the effect of betamethasone on maternal physiology [18].’

ABSTRACT

“The effects of betamethasone typically diminished within four days after initial administration” The meaning of this sentence is not clear. Data showed a significant difference between day 1 compared to day 0 or day 3, but a trend over the first 4 days is not clear.

Please see the updated abstract.

Line 26 – 28: ‘Within four days after initial administration of betamethasone, HR decreases and HRV features increase again, indicating a diminishing effect of betamethasone from a few days after administration.

METHODS

RESULTS

Fig 1: It might be useful to assess if the difference between D0 and D1 can be observed when the analysis is restricted to those who had a measurement at both instants.

Thank you for this suggestion. We have added this to the results. Please see lines 241 to 245:

To confirm the increase observed in HR from pre-betamethasone administration (day 0) to post-administration (day 1), we also considered a subanalysis with only participants who had measurements on both these days (n = 12). Correspondingly, we observed a similar increase from a median of 81.7 bpm (interquartile range: 77.8 - 83.5 bpm) to a median of 88.4 bpm (interquartile range: 85.0 - 91.8 bpm).

“all features linked to parasympathetic activity (RMSSD, pNN50, and HF) are significantly altered” – possibly it is clearer to describe the direction of the change = decreased.

Thank you for the suggestion. Please see the change made:

Line 252: ‘While no significant differences are seen in SDNN and LF, all features linked to parasympathetic activity (RMSSD, pNN50, and HF) are significantly decreased…

“The drop in SampEn……” : I thought that higher heart rate is generally associated with a reduction in variability of the signal. Are the changes that you observe larger than you would expect just from the HR shift?

Thank you for this observation, the increased HR could indeed be contributing to the decreased HRV. Please see the update made in the manuscript:

Line 319 - 324: ‘It should be noted that an increased HR typically results in lower variability in the HR signal, translating to generally lower HRV [36]. However, while the increased HR likely contributes to the decreased HRV features, SDNN – which is a measure of overall variability – is not significantly reduced. This suggests that additional physiological mechanisms likely also play a role in reducing the affected HRV features.’

DISCUSSION

You might describe more clearly that the difference of the numbers of registrations between D0 and D1 and following days could introduce bias. Ideally the study should have a repeated measurements design, but I fully understand that general clinical practice precludes this.

As per your suggestion under the RESULTS section, we have done a sub-analysis of how HR changes between day 0 and day 1 for participants who have measurements available for both days (lines 241 to 245). Furthermore, we have added the differing number of measurements per day to our limitations and stated the need for a future study with a paired analysis to confirm our results.

Lines 293 - 295:  ‘Still, more nuanced effects of betamethasone are perhaps obscured by … the differing number of measurements available per day …

Lines 389 - 391: ‘The next step would be to confirm these findings with a paired analysis from a prospective study dedicated to investigating the effect of betamethasone on maternal physiology [18].’

Line 331: There is some information on glucocortosteoids (but not on betamethasone) and ANS : e.g Clin Pharmacol Ther 1995;58:90-8, which shows (approximately) comparable data to this study

Thank you for this suggestion. We have incorporated this study into our discussion.

Lines 372 – 378: ‘While the stress response is typically associated with sympathetic activation, our results indicate that sympathetic activity is not significantly altered after betamethasone administration (LF from Figure 2). Conversely, one study has shown that administering exogenous glucocorticoids daily for a week inhibits baseline sympathetic outflow [48]. Although differences in study setup prohibit direct comparison between our results and theirs, it is evident that the relationship between HPA-axis and the ANS is still poorly understood…

“Future studies should further investigate the potential long-term impact of these effects, as women with complicated pregnancies already have abnormal autonomic regulation and an increased risk for cardiovascular disease later in life.” I doubt if it is realistic to expect a long-term effect of a single course of betamethasone in adult women. Furthermore, the increased risk of later cardiovascular disease depends on the same risk factors that cause the obstetric pathology.

Thank you for your comment. Please see the updated version in lines 391 to 395:

‘Future studies should investigate the potential impact of betamethasone on additional markers of maternal health, for example, BP. This is particularly important in pregnant women with already elevated BP, such as those with hypertensive disorders of pregnancy, as corticosteroids are likely to further increase BP [34].’